# Lettuce Biofortification with Selenium in Chitosan-Polyacrylic Acid Complexes

**Paola Leija-Martínez** [1], **Adalberto Benavides-Mendoza** [1], **Marcelino Cabrera-De La Fuente** [1], **Armando Robledo-Olivo** [2], **Hortensia Ortega-Ortíz** [3], **Alberto Sandoval-Rangel** [1] and **Susana González-Morales** [4,*]

1   Departamento de Horticultura, Universidad Autónoma Agraria Antonio Narro, Saltillo, Coahuila 25315, Mexico; pclm15@hotmail.com (P.L.-M.); abenmen@gmail.com (A.B.-M.); cafum7@yahoo.com (M.C.-D.L.F.); asandovalr16@gmail.com (A.S.-R.)
2   Departamento de Ciencia y Tecnología de los Alimentos, Universidad Autónoma Agraria Antonio Narro, Saltillo, Coahuila 25315, Mexico; armando.robledo@outlook.com
3   Centro de Investigación en Química Aplicada, Saltillo 25294, Mexico; hortensia.ortega@ciqa.edu.mx
4   CONACYT-Universidad Autónoma Agraria Antonio Narro, Saltillo 25315, Mexico
*   Correspondence: sgonzalezmo@conacyt.mx; Tel.: +52-844-411-0303

**Abstract:** Selenium (Se) is an essential element of the human diet. Therefore, it is necessary to implement Se in selenium-deficient soils and in the nutrient solution of soilless system culture. Although it is not considered as an essential element for plants, Se provides benefits at the level of redox metabolism, increasing the resistance of plants to various stress factors. The increase of the availability of Se, with the use of biopolymer complexes, was sought in *Lactuca sativa* var. Great Lakes, grown in substrate pots treated with $SeO_2$ (5 mg Se/plant), chitosan-polyacrylic acid complex + Se (Cs-PAA + Se) (5 mg Se/plant), and chitosan-polyacrylic acid complex (Cs-PAA). The redox metabolism was modified by increasing the enzymatic activity of catalase and glutathione peroxidase. The use of Cs-PAA + Se biopolymer complexes increased Se up to 24 mg/Kg dry weight (DW) in plant tissues.

**Keywords:** biofortification; antioxidants; soilless culture; nutraceutical quality; enzymatic activity; plant resistance

## 1. Introduction

Selenium is essential in the human diet, since a low intake has been linked to a weak immune system and cognitive decline [1]. On the contrary, the optimal intake of Se entails benefits, such as reducing the risk of different types of cancer, Alzheimer's disease, among others, and is necessary for thyroid function since the thyroid gland is the largest reservoir of Se in the human body, having different functions [2]. The adequate recommended intake of Se in humans has been proposed between 50–60 µg/day, on the contrary, an intake of 350–700 µg/day may result toxic [3]. It is known that Se is related to antioxidant metabolism through its role as a cofactor of selenoenzymes [4]. Therefore, its intake promotes the synthesis of antioxidant compounds, inducing changes in the cellular redox balance. The consumption of foods with high concentrations of antioxidants contributes to the protection of the cells against oxidative stress preventing some degenerative diseases. Free radicals cause oxidative chain reactions that can be neutralized through the action of antioxidant enzymes, such as catalase (CAT) and glutathione peroxidase (GPX). These enzymes increase their activity with the presence of Se in plants [5]. Several studies carried out on strawberry [6], tomato [7,8], and corn salad [9] crops have managed to biofortify the organs of consumption with favorable results with

respect to plant growth and antioxidant content. It has been reported that an application of 5 mg per plant in the whole production cycle is effective to achieve biofortification in tomato fruits [5].

Since Se is a non-renewable resource, it is necessary to incorporate it into agricultural fertilization in an appropriate manner, in Se-deficient soils and in hydroponic and soilless culture systems, in order to avoid wastage of the element. The use of biopolymers to encapsulate elements in order to improve their absorption by plants is a technique that can be effective for crop biofortification with Se. Biopolymers, such as chitosan (Cs) and polyacrylic acid (PAA) are capable of encapsulating active ingredients using an aqueous system at room temperature [10]. The mixture between these polymers has physicochemical properties and relatively simple processing techniques that come from synthetic polymers and an adequate biocompatibility from natural polymers, this confers varied applications, such as the immobilization and prolonged release of various chemical elements or bioactive agents. These biopolymer complexes may be used in order to avoid the loss of Se through leaching, adsorption to organic components of the substrate, and volatilization by microorganisms [11]. In the present work, it was sought to biofortify lettuce plants with 5 mg Se/plant ($SeO_2$) by applying it alone and absorbed in a complex with biopolymers, studying the impact on the plant growth, the antioxidant activity, and the accumulation of Se.

## 2. Materials and Methods

### 2.1. Preparation of Non-Stoichiometric Interpolyelectrolyte Complexes (NPEC) of Chitosan-Polyacrylic Acid (Cs-PAA) and Cs-PAA + Se

Chitosan (Cs) was supplied by Marine Chemicals (Meron, Kerala, India). Deacetylation degree was 99%, molecular weight Mv = 200,000 g/mol and was determined by intrinsic viscosity in an Ubbelohde viscometer by the ASTM D2857 method using a mixture of acetic acid/sodium acetate solutions at 30 °C applying Mark-Houwink equation: $\eta = KMv^{\alpha}$, where k = 0.076 and $\alpha$ = 0.76. Poly (acrylic acid) molecular weight Mw = 450,000 and selenium oxide ($SeO_2$) were acquired from Aldrich. Water-soluble NPEC were prepared at the Center for Research in Applied Chemistry with polycation (Cs) and polyanion (PAA) [12]. The NPEC (Cs-PAA) with the composition $\varphi$ = [CS]/ [PAA] = 3 was used. The square brackets denote molar concentrations of polyions prepared by mixing water solutions of Cs and PAA at corresponding quantities of the polyelectrolytes solutions at pH 2, and then adjusting the solution with phosphate buffer at pH 7.4. Lastly, from $SeO_2$, 50 mg Se/L of complex was added, which is the amount needed to make the applications of Cs-PAA + Se.

### 2.2. Plant Material and Treatments

A soilless lettuce (*Lactuca sativa* L.) var. Great Lakes (FAX Seeds, Jalisco, Mexico) crop was established in a greenhouse with polyethylene cover, 50%–60% relative humidity, and an average temperature of 28 °C. Lettuce seeds were sown in black polyethylene trays filled with sphagnum peat moss and perlite in a 1:1 ratio *v/v*. At 40 days after sowing, seedlings were transplanted to black polyethylene containers filled with 5 L of substrate mix of sphagnum peat moss and perlite. Steiner [13] nutrient solution was applied by a single pass drip irrigation system, with a pH (6–6.5) adjusted daily with phosphoric acid. The treatments application started seven days after transplant (DAT). Each treatment was composed of 20 lettuce plants, placed randomly in four lines. The treatments were applied manually to the substrate. A treatment with $SeO_2$ (50 mg/L Se) was applied with a volume of 12.5 ml of solution in each application. The next treatment consisted in applying the Se (50 mg/L) in Cs-PAA complex with the same volume as in the previous treatment. In both treatments, a total of 5 mg of Se per plant was applied, divided into eight weekly applications. On the other hand, the Cs-PAA without Se was also applied, and a control with no application.

*2.3. Sampling*

Lettuces were harvested at 60 DAT, measured and weighted to determinate total growth and total crop yield. At the same time, samples were collected to quantify the biochemical variables and Se content, obtaining five samples by each treatment and control.

*2.4. Yield and Biomass Production*

In order to determine total crop yield, the lettuce heads were harvested and weighed on an analytical scale (Ohaus Corporation, Pine Brook, NJ, USA) to determine the total fresh weight (FW). In order to determine the total accumulated biomass, five plants of each treatment were dehydrated in an oven at 80 °C for 48 h. Once total dehydrated, the dry weight was registered.

*2.5. Biochemical Analysis*

In order to determinate nutraceutical quality, five samples were frozen at $-20$ °C during 48 h then freeze-dry in a Freeze Dryer (Labconco, Freezone 6, Kansas City, MO, USA) at $133 \times 10^{-3}$ mbar and $-80$°C during 48 h. Once completely dried, samples were pulverized with a porcelain mortar. For the biomolecules extraction, 200 mg of powdered sample was placed in a 2 mL tube and 20 mg of polyvinylpyrrolidone (PVP) (Sigma-Aldrich Corporation, Saint Louis, MO, USA) was added to stabilize the enzymes; 1.5 mL of phosphate buffer 0.1 M (pH 7–7.2) was added, homogenized by vortex for 20 seconds each sample, sonicated during five minutes, and centrifuged at 12,500 rpm at 4 °C for 10 minutes, collecting the supernatant in order to perform the analysis.

2.5.1. Total Proteins

The Bradford spectrophotometric technique [14] was used for protein quantification. One hundred microliters of the protein extract was placed in an assay tube, and adding 5 mL of the Bradford reagent, and let to stand for five minutes. Once the incubation time passed, the absorbance was read at a wavelength of 595 nm with a UV-Vis spectrophotometer (Thermo Scientific Model G10S, Waltham, MA, USA) The results were registered and extrapolated to a calibration curve of bovine serum albumin (BSA) (Sigma-Aldrich Corporation, Saint Louis, MO, USA), reporting the results in m/g.

2.5.2. Catalase Activity (CAT) (EQ 1.11.1.6)

The enzymatic activity of catalase was quantified by a spectrophotometric technique, with two reaction times. The assay mixture consisted on 100 μL of protein extract and 900 μL of 100 mM $H_2O_2$ (CTR Scientific, Monterrey, Nuevo León, Mexico), and was added 400 μL of 5% $H_2SO_4$ (Sigma-Aldrich Corporation, Saint Louis, MO, USA) to stop the reaction, this was carried out under stirring at 24 °C. These assays were read with a spectrophotometer (Thermo Scientific Model G10S, Waltham, MA, USA) at 270 nm; time reaction zero (T0) was recorded. For time reaction one (T1), the mixture of protein extract and 100 mM $H_2O_2$ was stirred at 24 °C, after 1 min 400 μL 5% of $H_2SO_4$ was added to stop the enzymatic activity. The remaining $H_2O_2$ was read with a spectrophotometer at 270 nm. For this analysis, a blank for each sample was used, consisting of 100 μL of biomolecules extract, 900 μL of 0.1 M phosphate buffer and 400 μL of 5% $H_2SO_4$. Units of catalase activity were expressed as mM $H_2O_2$ min$^{-1}$/ total proteins [15].

2.5.3. Glutathione Peroxidase Activity (GPX) (EQ 1.11.1.9)

A spectrophotometric method [16] was used with $H_2O_2$ as substrate; 200 μL of biomolecules extract was placed in a test tube plus 400 μL of reduced glutathione 0.1 M and 200 μL $Na_2HPO_4$ (Fermont, Monterrey, Nuevo León, Mexico) 0.067 M. This mixture was preheated in a water bath at 25 °C for 5 min, then 200 μL of 1.3 mM $H_2O_2$ was added to start the catalytic reaction. The reaction lasted 10 min and was terminated by adding 1 mL 1% trichloric acetic acid (Sigma-Aldrich Corporation, Saint Louis, MO, USA), and the mixture was put into an ice bath for 30 min. The assay mixture was

centrifuged for 10 min at 3000 rpm; 480 μL of the supernatant was placed into a cuvette, with 2.2 mL of 0.32 M $Na_2HPO_4$ and 320 μL of 1.0 mM 5-5′-dithiobis 2-nitrobenzoic acid (DTNB, Sigma-Aldrich Corporation, Saint Louis, MO, USA) was added for color development. The enzyme activity was determined as a decrease in GSH within the reaction time, expressed as mg/L GSH min/total proteins.

### 2.5.4. Glutathione (GSH)

Glutathione was quantified following a spectrophotometric technique [16], by reaction with the 5,5-dithio-bis-2-nitro benzoic acid (DTNB). Four hundred and eighty microliters of protein extract was placed in a tube then added 2.2 mL of 0.32 M $Na_2HPO_4$ plus 320 μL of 1.0 mM DTNB dye. The assay was mixed and read on a spectrophotometer (Thermo Scientific Model G10S, Waltham, MA, USA) at 412 nm. The units were reported in mg glutathione/g dry tissue.

### 2.5.5. Total Phenols Content

Regarding the total phenolic extraction, five samples of each treatment were used. Two hundred milligrams of a lyophilized and macerated sample was taken, and 1 ml of water-acetone 1:1 solution was added, centrifuged at 10,000 rpm at 4 °C for 10 min, and the supernatant was collected to initiate the reaction. The assays were prepared using 200 μL of Folin-Ciocalteu 1 M reagent, 50 μL of the extract, 500 μL of 20% $Na_2CO_3$, 1 mL water-acetone 1:1 mix, and 5 mL of distillated water. Blank was composed by the same components adding 50 μL of water-acetone mix instead of the extract. Subsequently, assays were let to incubate at 45 °C for 30 min. The samples were read with a spectrophotometer (Thermo Scientific Model G10S, Waltham, MA, USA) at a wavelength of 750 nm and the results were recorded in μg/g [17].

### *2.6. Selenium Content*

Selenium was extracted using a wet digestion technique [18]. For digestion, 500 mg of the dehydrated sample was placed in a beaker with 30 mL of nitric acid, and heated during two hours in a heating plate, until clarification of the mixture. Finally, the sample was collected and taken to a volume of 50 mL with deionized water and filtered with Whaltman # 42 filter paper. Mineral content was read in an inductively coupled plasma optical emission spectrometer (ICP-OES, Thermo Scientific Jarrell Ash, Model iCAP 7000 Series, Waltham MA, USA). The calibration curve was elaborated with a Selenium standard (High Purity Standars, North Charleston, SC, USA), with 5 points from 0.005 to 2 ppm. The plasma was generated with a torch at a temperature of 10000 ° K by the ionization of argon.

### *2.7. Statistical Analysis*

The experimental design was completely randomized with five replicates per treatment on each variable, with one plant considered as an experimental unit. Infostat software (Grupo InfoStat, Córdoba, Argentina) was used, in which a Fisher Least Significant Difference test ($\alpha \leq 0.05$) was performed for all variables.

## 3. Results and Discussion

### *3.1. Yield and Biomass Production*

The treatments applied did not exert an effect on the variables related to yield and biomass, (Table 1). The analysis of accumulated biomass in the form of the dry weight of plant tissues is a useful indicator in order to determine the toxicity of an element. Based on the results obtained in this experiment, in which there was no significant difference between the treatments and the control, it is suggested that the applied concentrations of Se were nontoxic. The application of the Cs-PAA complexes can also be considered safe for the development of the crop. In a crop of radish plants established in soil treated with 40 μM $SeO_4^-$ per plant, the authors [19] reported a decrease in dry

weight of leaves and roots by about 35% and 18% respectively. On the other hand, when cultivating the radish plants in a soilless culture, they did not find significant differences in the dry weight of the leaves.

### 3.2. Biochemical Variables

According to the results analyzed, the content of proteins, phenolic compounds, and glutathione, are not affected by the application of the treatments in comparison with the control, as it is detailed in Table 1.

**Table 1.** Comparison of means of biomass and biochemical variables.

| Treatment | Biomass (g DW) | PROT (mg/g) | GSH (mg/g) | PHEN (µg/g) |
|---|---|---|---|---|
| Control | 51.35 ± 3.19 [a¥] | 2.24 ± 0.56 [a] | 0.81 ± 0.02 [a] | 63.51 ± 7.49 [ab] |
| SeO$_2$ | 51.67 ± 2.43 [a] | 2.87 ± 0.51 [a] | 0.83 ± 0.01 [a] | 88.4 ± 14.05 [a] |
| Cs-PAA + SeO$_2$ | 47.27 ± 4.12 [a] | 2.05 ± 0.35 [a] | 0.95 ± 0.38 [a] | 61.1 ± 11.11 [ab] |
| Cs-PAA | 50.17 ± 2.41 [a] | 3.57 ± 0.82 [a] | 0.97 ± 0.58 [a] | 48.76 ± 6.81 [b] |

[¥] Means with the same letter are statistically equal (LSD, $p \leq 0.05$). Mean ± standard error of the mean ($n = 5$). PROT, proteins; CAT, catalase activity; GSH, glutathione; PHEN, total phenolic compounds.

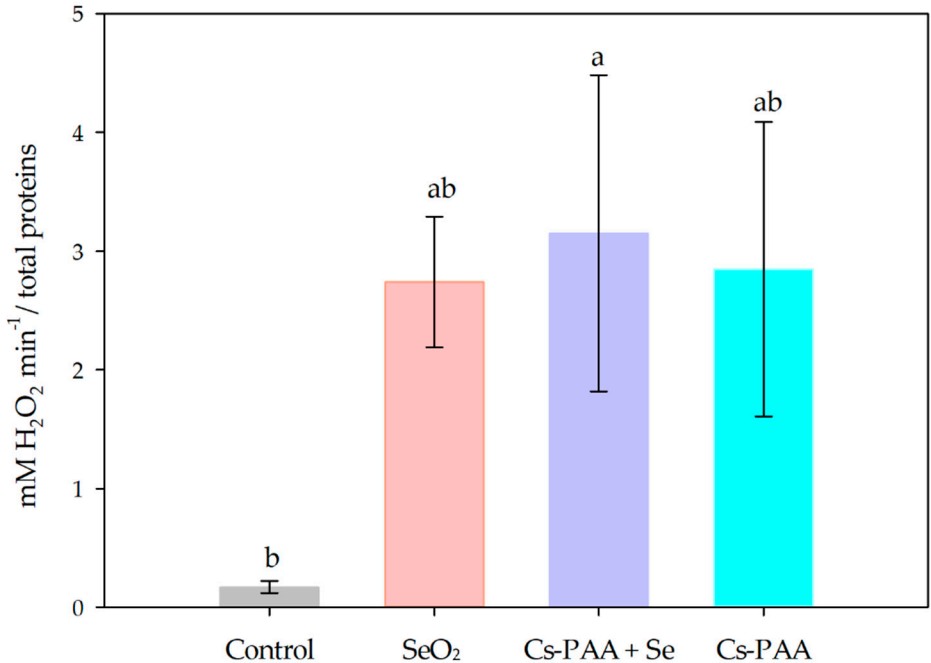

**Figure 1.** Means comparison of the enzymatic activity of catalase. Means with the same letter are statistically equal (LSD, $p \leq 0.05$). Bars represent standard error.

In the enzymatic activity of catalase (Figure 1), SeO$_2$ and Cs-PAA treatments were not able to surpass the control, however, when applied combined in the treatment of Cs-PAA + Se, was induced a greater enzymatic activity compared to the control. In another study with spirulina (*Spirulina platensis*) [20], Se applied at 150 mg/L or less, induced an increase in the activity of peroxidase enzymes, including catalase. Similarly, the application of a higher concentration (>175 mg/L) induced enzymatic activity, but in conjunction with an increase in lipid peroxidation and a decrease in biomass and photosynthetic pigment content. Some studies [21] suggest that Se also induces an increase in the enzymatic activity of CAT when used in concentrations of 5 to 10 mg/L to reduce oxidative stress in plants of *Triticum aestivum* L. under drought stress conditions.

In the statistical analysis of the GPX activity, significant differences were found (Figure 2). All the treatments applied had higher values of GPX activity compared to the control. Both the $SeO_2$ and the Cs-PAA complex applied by themselves increase GPX activity, (729% and 789% respectively); however, when applied together as Cs-PAA + $SeO_2$ a higher enzymatic activity is obtained (1031% increase over control). The above results agreed with those of a study in ryegrass [22], whose authors reported an increase in GPX activity by applying Se at 1 mg/Kg. Similarly, other works [23] also reported an increase in GPX activity with applications of different Se concentrations up to 150 mg/L, but in conjunction with a reduction of biomass and chlorophyll when applying more than 175 mg/L in *Spirulina platensis*. Since all the treatments applied in the present work induced an increase in the GPX activity, it is suggested that both the Se and the Cs-PAA complex induce this increase by themselves, and when applied as a whole the effect is enhanced.

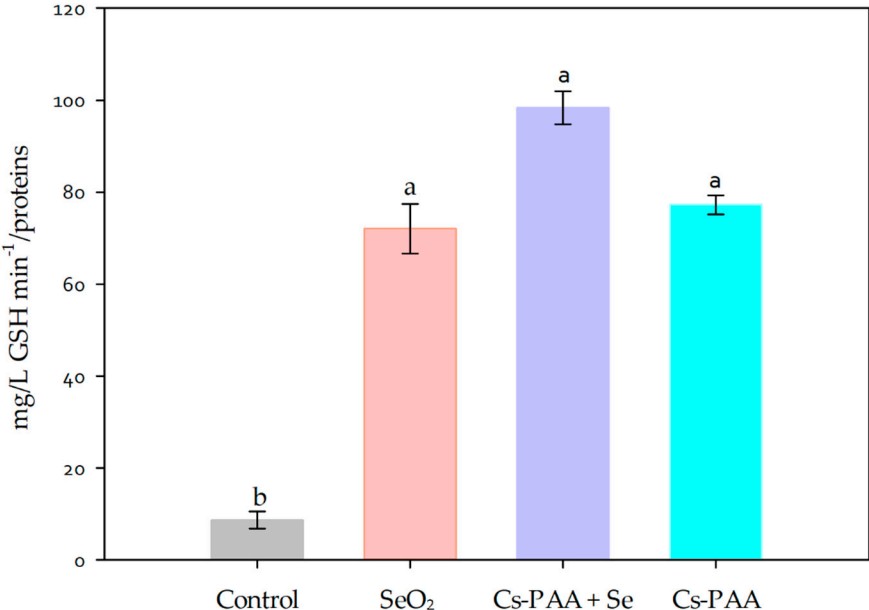

**Figure 2.** Means comparison of the enzymatic activity of glutathione peroxidase. Means with the same letter are statistically equal (LSD, $p \leq 0.05$). Bars represent standard error.

GSH content was not affected by the treatments, in the same way as in an experiment with Chinese cabbage subjected to stress, due to the high concentration of heavy metal Cd, where GSH increase when Se was applied in combination with Si; however, there was no difference when applying Se alone [24]. In contrast, in an experiment performed on wheat [25], GSH increased by 18% when Se was applied in conjunction with S.

As in the present work, in another study [19] with radish plants using foliar applications of up to 20 mg Se per plant, no differences were found compared to the control in phenolic compounds. In contrast, in an experiment under field conditions of *Allium cepa* L. Se applied 50 µg/mL$^{-1}$ increased the content of total phenolic compounds compared to the control [26].

The evaluation of enzymatic activity of antioxidant enzymes, as well as the content of non-enzymatic antioxidants is an indicator that can give a clear notion about the redox metabolism of a plant, which in turn can be translated as the nutraceutical quality of a crop by its antioxidant content. Based on this fact, the effect of Se on enzymatic and non-enzymatic antioxidants depends widely on the plant species used.

### 3.3. Selenium Biofortification

The treatment of $SeO_2$ increased Se content 236% over control, and the treatment of Cs-PAA + Se had an additional increase of 74% over the above (Figure 3). There was no significant difference in the

accumulation of Se between the treatments of $SeO_2$ and Cs-PAA + Se. It is possible that a difference will occur with a previous significant degradation of the polymer, which may happen in a second growing season using the same substrate. Therefore, it is feasible to use Cs-PAA + $SeO_2$ for biofortification.

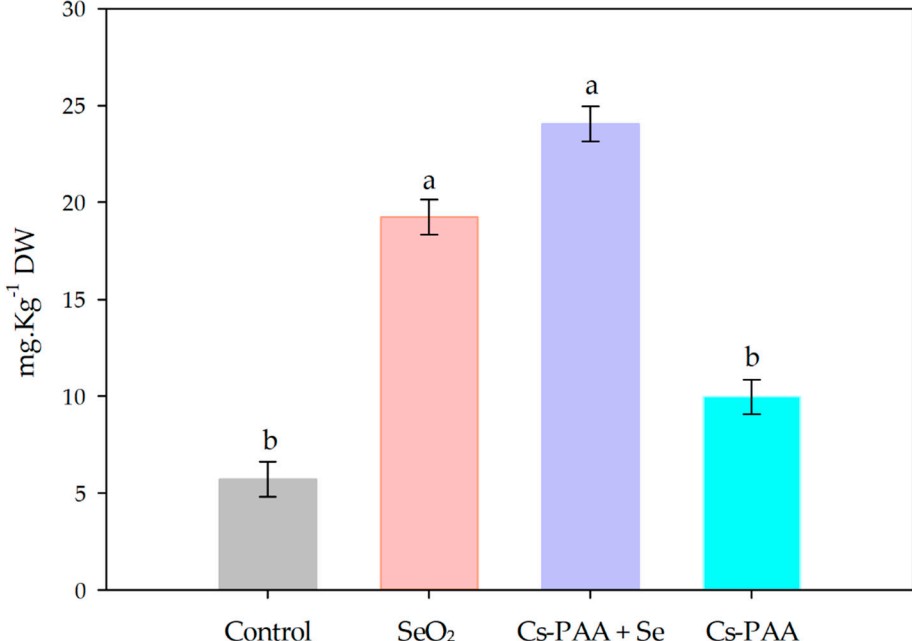

**Figure 3.** Mean values of the Se accumulation in lettuce plants. Means with the same letter are statistically equal (LSD, $p \leq 0.05$). Bars represent standard error.

The results of the biofortification techniques of Se have a variable behavior according to the plant species, the form of application, dose and concentration of Se, and the production system implemented. For instance, in a soilless crop of lettuce [27] with applications of $SeO_3$ and $SeO_4$ in concentrations from 2 to 64 µmol/L in the nutrient solution, the authors reported that at the highest concentrations Se accumulation reached up to ~22 mg/Kg DW however it also resulted in Se toxicity with a decrease in biomass. Similarly, in lettuce seedlings [16] with $H_2SeO_4$ applied to the soil from 0.1 to 1 mg/Kg there was an increase in Se accumulation, but it was toxic at the highest concentration; it accumulated up to 270 mg/Kg DW in the tissues but reducing the biomass up to 66% compared to the control. On the other hand, other studies [28] suggest the application of $SeO_4^{2-}$ in conjunction with $IO_3^-$ in order to achieve a double biofortification of lettuce "Melodion" cv. in hydroponic cultivation. There is a synergistic interaction between both compounds applied in foliar applications, suggesting a transport through the phloem tissues.

The values of Se concentration obtained in this study seem adequate from the perspective of biofortification, since with the application of Cs-PAA + $SeO_2$ it was possible to biofortify Great Lakes lettuce, up to 24 mg/Kg DW. Considering a portion of lettuce of 100 g FW, this adds up to 165 µg Se per serving of lettuce, which is an adequate amount for daily consumption since it exceeds the minimum requirement, but does not reach the doses that, under certain circumstances, could become harmful to the consumer. The recommended daily intake of Se in the human diet varies by region, in addition to the consumer particular health status. According to U.S. Food and Nutrition Board, the recommended intake is 60 to 75 µg Se daily [29]. However, in order to obtain the benefits that result from the intake of Se, such as reducing cancer risk, enhancing male fertility, and generally improving immune responses, it is necessary to increase this daily dose up to 200–300 µg daily [1]. Nevertheless, there are mixed reports about the recommended intake, since according to the Nutritional Prevention of Cancer (NPC), they reported that the consumption of 200 µg a day in yeast supplements decreases the incidence only in people with low levels of Se in plasma, causing the opposite effect in people

with adequate levels of Se [30]. Based on this fact, special attention should be paid to the amounts of accumulated Se by biofortified crops, due to the small threshold of appropriate dose.

## 4. Conclusions

Our results indicate that Cs-PAA complexes can be beneficial in biofortification processes, due to their tendency to increase the absorption of Se, as well as generating an increase in the activity of CAT and GPX antioxidant enzyme, without affecting the development of the crop. Based on what has been observed, Se and the Cs-PAA complexes were effective to enhance plant resistances, as well as the nutraceutical quality of Great Lakes lettuce.

**Author Contributions:** S.G.-M. conceived and designed the experiments; P.L.-M. performed the analysis of laboratory and field experiments; A.B.-M., M.C.-D., and A.S.-R. performed the data analysis; H.O.-O. and A.R.-O. contributed reagents, materials, and investigation. All authors were responsible for manuscript writing. All authors read and approved the final manuscript.

**Funding:** This research received no external funding.

**Acknowledgments:** Special thanks to LCQ Jesús Alejando Espinosa Muñoz for his support in the determination of selenium by ICP-OES. UAAAN Proyecto Interno 38111-425102001-2162: "Calidad Nutracéutica y Expresión Génica de Dos Especies Hortícolas Biofortificadas con Selenio Iónico Absorbido en Complejos de Poliácido Acrílico-Quitosán".

**Conflicts of Interest:** The authors declare no conflict of interest.

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
