# Peer review of "Lettuce Biofortification with Selenium in Chitosan-Polyacrylic Acid Complexes"

_agronomy, doi:10.3390/agronomy8120275_

Round 1

Reviewer 1 Report

The ms submitted by Leija-Martínez et al. deals with an experiment of biofortification of lettuce plants with the application of selenium in chitosan-polyacrylic complexes. the authors evaluates both the yield and some biochemical parameters of treated plants as well as their selenium content. The topic covered by the authors is not really novel and the authors did not use very innovative approaches, however the ms is overall suitable for the journal. Indeed, I suggest a careful review of the english by a native speaker and a particular improvement of the results and discussion section. Based on my comments and suggestions below, I recommend the ms to be accepted after major revision.

L30: please delete "on its functioning"

L40: please add here a short paragraph introducing some experiences of biofortification in different plant species before describing directly the use of biopolymers. I suggest to cite these works: DOI: 10.1021/jf4031822; DOI: 10.1071/CP14218; DOI: 10.3389/fpls.2017.01887

L56: please add "and" after "g/mol"

L68: please add "," after "cover"

L71: please add "." after "sown"

L72: please add the pH of the solution

L76: Table 1 is not necessary and can be included in the text

L81: please delete "and control". Control can be considered one of the treatments.

L94: please add the molarity of phosphate buffer

L115: molarity of reagents is missing

L120: please move "was added" after "H2O2"

L125: please write "determined" instead of "determinate"

L130-131: please rephrase this sentence.

L134: please delete "and control"

L141: this table is not necessary and the description of the method can be included in the text.

L147: please write "the sample" instead of "it"

L149: maybe you refer to ICP-OES? if so write the name extensively instead of writing "plasma emission spectrophotometer"

L157-159: please delete

L179-181: please rephrase, it is not clear

L183: please write "Similarly, the application of a higher..."

L201: are the error bars in the graph standard error or standard deviation? please specify

L206: any other experiences to cite?

L219-223: please rephrase, it is not clear

L224: please use the same name for the treatments as in Figure 1 ( Cs-PAA+Se, Cs-PAA)

L232: please add "applied" before "to the soil"

L237-238: please rephrase

L238: please write only Se instead of selenium. please check the whole ms

L240: please add "." after "DW"

please check the references since many errors are present.

Author Response

Reviewer 1.

L30: please delete "on its functioning"

Deleted in L33

L40: please add here a short paragraph introducing some experiences of biofortification in different plant species before describing directly the use of biopolymers. I suggest to cite these works: DOI: 10.1021/jf4031822; DOI: 10.1071/CP14218; DOI: 10.3389/fpls.2017.01887

Information added in L42 to L45.

L56: please add "and" after "g/mol"

Added in L64.

L68: please add "," after "cover"

Added in L76.

L71: please add "." after "sown"

Sentence was changed in L79

L72: please add the pH of the solution

Added in L80.

L76: Table 1 is not necessary and can be included in the text

Table 1 was removed.

L81: please delete "and control". Control can be considered one of the treatments.

Deleted.

L94: please add the molarity of phosphate buffer

Added in L103.

L115: molarity of reagents is missing

Added in L122 and L125.

L120: please move "was added" after "H2O2"

Moved as indicated in L130.

L125: please write "determined" instead of "determinate"

Changed in L135.

L130-131: please rephrase this sentence.

Changed in L140.

L134: please delete "and control"

Deleted in L144.

L141: this table is not necessary and the description of the method can be included in the text.

Table 2 deleted.

L147: please write "the sample" instead of "it"

Changed in L157.

L149: maybe you refer to ICP-OES? if so write the name extensively instead of writing "plasma emission spectrophotometer"

Details of the equipment were added in L159-L162.

L157-159: please delete

Deleted.

L179-181: please rephrase, it is not clear

Phrase was deleted.

L183: please write "Similarly, the application of a higher..."

Changed in L195.

L201: are the error bars in the graph standard error or standard deviation? please specify

The bars represent standard error. The specification was added at each figure description.

L206: any other experiences to cite?

Added in L218-L219.

L219-223: please rephrase, it is not clear

Changed in L232-L236.

L224: please use the same name for the treatments as in Figure 1 ( Cs-PAA+Se, Cs-PAA)

Changed in figure caption.

L232: please add "applied" before "to the soil"

Changed in L245.

L237-238: please rephrase

Changed in L248-L251.

L238: please write only Se instead of selenium. please check the whole ms

Changes were made throughout the manuscript.

L240: please add "." after "DW"

Changed in L254.

please check the references since many errors are present.

References were corrected according to the journal guidelines.

Reviewer 2 Report

Abstract should be rewritten, Latina name of lettuce, no shortcuts without explanation.

Introduction lacks of information about Se biofortification. Why authors choose such concentration?

Units should be corrected and written in one form mg/l or mg.l-1). Same for numbers (line 64 and 95).

Why are some results presented in a table and others in a graph?

Statistical analysis should be changed. Results showed big diferences that seemed significant.

Results described in lines 218-221 did not correspond with figure 2!

Author Response

Reviewer 2.

Abstract should be rewritten, Latina name of lettuce, no shortcuts without explanation.

Latin name and abbreviations specification was added.

Introduction lacks of information about Se biofortification. Why authors choose such concentration?

Added brief information on biofortification and amount of selenium recommended to apply to the crop in L42-L45. Fellow researchers have applied 5 mg Se per plant obtaining favorable results, therefore we decided to use the same amount of Se.

Units should be corrected and written in one form mg/l or mg.l-1). Same for numbers (line 64 and 95).

Changes were made throughout the manuscript.

Why are some results presented in a table and others in a graph?

In order not to extend the manuscript unnecessarily, it was decided to graphically illustrate only the relevant results.

Statistical analysis should be changed. Results showed big diferences that seemed significant.

We analyzed the results with an ANOVA, which did not express significant differences since the standard error is very broad. However, we performed the LSD test (α = 0.05) and differences were found in the variables of CAT activity and phenolic compounds. After making a literature review, we concluded in reporting the results of the LSD test. Therefore, the results table was changed and a graph was added to illustrate the CAT results.

Results described in lines 218-221 did not correspond with figure 2!

The label of the figure was corrected.

Reviewer 3 Report

This paper describes an experiment that shows that selenium concentrations in lettuce can be increased by the use of biopolymer encapsulated Se without detrimental impacts on yield.
The methods appear to be sound. It is reasonably well written but some of the grammar is confusing so it would benefit from editing.
There appears to be some confusion about the discussion of the resuts in fig 2. The figure suggests that the Se content of lettuce supplemented with Cs-PAA and Cs-PAA+Se were significantly higher than the plants supplemented with SeO2 bu the text states that there was no difference.  I suspect that some of these results are mis-labeled because Cs-PAA alone should not increase Se more than SeO2 application. This needs to clarified. If the data has not been mislabeled then it needs to be explained.

Author Response

Reviewer 3.

This paper describes an experiment that shows that selenium concentrations in lettuce can be increased by the use of biopolymer encapsulated Se without detrimental impacts on yield.

The methods appear to be sound. It is reasonably well written but some of the grammar is confusing so it would benefit from editing.

There appears to be some confusion about the discussion of the resuts in fig 2. The figure suggests that the Se content of lettuce supplemented with Cs-PAA and Cs-PAA+Se were significantly higher than the plants supplemented with SeO2 bu the text states that there was no difference.  I suspect that some of these results are mis-labeled because Cs-PAA alone should not increase Se more than SeO2 application. This needs to clarified. If the data has not been mislabeled, then it needs to be explained.

The graphic was mislabeled; it has now been corrected.

L14 only where it is lacking - can be toxic at high concs

Changed in L17

L18 provide definition

Added on L21-L22

L21 increased

Changed in L24

L38 New paragraph

Changed in L46

L39 Only in deficient soils . need to take care to avoid high concns

Changed in L47

L47 uncommon word. sugest changing to alternative such as leaching, or increasing solubility

Changed in L56

L67 hidroponics?

The crop was not established in soil, however it was not established in water but in substrate, therefore we decided to consider it only as soilless.

L70 sowing

Changed in L78

L71 were nutrients recycled or single pass?

Information added in L80

L72 what was th ph set point?

Added in L80.

L74 what concentrations? and how was the system set up? was there a separate hydroponic system for each treatment How many plants were in each tray?

Information added in L82-L87.

L101 protein?

Changed in L111.

L118 ??? is this the same as the protein extract?

Changed in L128

L129 wording

Changed in L139

L137 does this need to be in a table? all reagents are the same. The only difference is that the blank has no extract added. Also - was the volume made up with water in the blank to ensure that the final reagent concs in the samples and blanks were the same?

Table 2 was removed, details of the assay were described in L147-L149

L148 what were the operating conditions (column type, carrier solution, temps etc). How was the standard curve constructed? Was a reference sample run? if so, what was it?

Details were added in L159-L162.

L157 delete

Deleted

L179 This sentence does not make sense. consider revising.

Changed in L191

L182 induced

Changed in L194

L183 peroxidase

Changed in L194

L184 induced

Changed in L195

L219 The PAA-CS_Se treatment in Fig 2 appears to be signficantly higher than the Se)2 treatment. Is this treatment different from teh Sc-PAA+SeO2 mentioned here?

The figure was mislabeled; it has been corrected in the new version of the manuscript.

L238 New paragraph here.

Changed in L252

Round 2

Reviewer 1 Report

The manuscript has been improved compared to the first version and it is now suitable for publication.

Reviewer 3 Report

I am satisfied with the changes that the authors have made. I think I is now acceptable for publication.